# Botanical Control of Citrus Green Mold and Peach Brown Rot on Fruits Assays Using a *Persicaria acuminata* Phytochemically Characterized Extract

**DOI:** 10.3390/plants10030425

**Published:** 2021-02-24

**Authors:** Melina G. Di Liberto, Gisela M. Seimandi, Laura N. Fernández, Verónica E. Ruiz, Laura A. Svetaz, Marcos G. Derita

**Affiliations:** 1Farmacognosia, Facultad de Ciencias Bioquímicas y Farmacéuticas, Universidad Nacional de Rosario, Suipacha 531, Rosario S2002LRK, Argentina; mdiliber@fbioyf.unr.edu.ar; 2ICiAgro Litoral, Facultad de Ciencias Agrarias, Universidad Nacional del Litoral, CONICET, Kreder 2805, Esperanza 3080HOF, Argentina; giselaseimandi@hotmail.com.ar (G.M.S.); laurafernandez1@gmail.com (L.N.F.)

**Keywords:** phytopathogenic fungi, *Persicaria acuminata*, antifungal, fruits, orange, peach, rot, sesquiterpenes

## Abstract

*Persicaria acuminata* (Polygonaceae) is a perennial herb that grows in the central area of Argentina and it is commonly used by native populations to heal infected wounds and other conditions related to fungal infections. In this article, we explored the in vitro antifungal activity of its ethyl acetate extract against a panel of three fruit phytopathogenic fungi including: *Penicillium digitatum*, *P. italicum,* and *Monilinia fructicola*. The sesquiterpenes isolated from the extract were also evaluated against these strains, demonstrating that the dialdehyde polygodial was the responsible for this activity. In order to encourage the use of the extract rather than the pure compound, we displayed ex vivo assays using fresh oranges and peaches inoculated with *P. digitatum* and *M. fructicola*, respectively, and subsequently treated by immersion with an extract solution of 250 and 62.5 µg/mL, respectively. There were no statistically significant differences between the treatments with commercial fungicides and the extract over the control of both fruit rots. The concentration of the active compound present in the extract used on fruit experiments was determined by Gas Chromatography-Mass Spectroscopy. Finally, cytotoxicity evaluation against Huh7 cells showed that *P. acuminata* extract was less cytotoxic than the commercial fungicides at the assayed concentrations. After these findings we could conclude that a chemically characterized extract of *P. acuminata* should be further developed to treat fungal diseases in fruits from an agro-ecological model.

## 1. Introduction

Plant diseases caused by phytopathogenic fungi are responsible for economic losses arising mainly from crop yield reduction, but also resulting from diminished product quality and safety; sometimes they also represent a risk for human and animal health due to food contamination and the accumulation of toxic residues in the environment. Due to market globalization and climate change, the problem is growing at an accelerated pace [1]. Since regulations on the use of new and existing fungicides are getting stricter, there is an urgent need to find and develop new chemical entities with antifungal activities [2]. Different naturally occurring compounds [3], semisynthetic derivatives [4], chitosan-based formulations [5], and plant products, including extracts [6] or essential oils [7], have been reported as part of this strategy. Moreover, some natural products and derivatives have been synthesized with the purpose of enlarging the offer of fungicides to be used on fruits [8].

*Penicillium digitatum* (Pers.) Sacc causes green mold rot disease, which is the most important post-harvest disease of citrus fruit worldwide. In predisposing conditions, losses may reach up to 90% of the crop [9]. Management of green mold is currently based on an integration of actions, such as minimizing fruit injury, sanitary practices, and treatments with fungicides [10]. The high number of pre- and post-harvest application of chemical fungicides has caused the development of *P. digitatum* resistant strains to several chemical groups [11]. Therefore, the requirement for alternative control strategies is increasing. The control of green mold without the application of chemical fungicides has been recently reviewed [11] and among these non-chemical treatments, natural compounds, irradiations, hot water treatments, salts, and biocontrol agents constitute promising strategies for *P. digitatum* management [10,11].

*Penicillium italicum* Wehmer causes blue mold rot disease, and this pathogen is a major threat as it grows faster than *P. digitatum* at the same environmental conditions. Blue mold may also predominate in fruits treated with benzimidazoles, since resistance to these fungicides occurs more frequently in isolates of *P. italicum* compared to *P. digitatum* [11].

*Monilinia fructicola* (G. Wint.) Honey is the causal agent of brown rot, a destructive pathogen on stone fruits worldwide, which also causes blossom blight and twig cankers [12,13]. In peaches, *M. fructicola* is responsible for the main fruit losses during the growing season and the post-harvest stage, being reported in almost all producing regions worldwide [14]. The management of the disease integrates the application of fungicides with cultural handling, such as the removal of mummified fruits and the pruning of twigs with cankers to reduce inoculum levels. At present, the management of *M. fructicola* with fungicides constitutes a challenge for several reasons [15]. A high number of applications are required to protect flowers and fruits while taking into account the long susceptibility period for the infection. Fungicides such as dithiocarbamates, which present low possibilities to generate resistance, have long waiting periods that inhibit its application at fruit maturity. Other fungicides with shorter waiting periods, such as dicarboxamides, benzimidazoles, or demethylation inhibitors, have shown high risk of resistance build-up [15].

*Persicaria acuminata* (Kunth) M.Gómez *syn*. *Polygonum acuminatum* formerly belonged to *Persicaria* section of *Polygonum* genus, but since 2017 it is accepted named is *P. acuminata* [16]. This is a native herb that grows in Argentina whose vernacular name is “catay grande” and it is used in Traditional Medicine as an antiseptic [17]. A few studies regarding its biological activities had been published up to 2009, highlighting its insecticidal activity against *Lutzomyia longipalis* [18]. Moreover, phytochemical studies and antifungal activities against human pathogenic fungi for this species were carried out in our research group from 2010 [19,20,21], but this is the first report for its phytopathogenic fungi control. The aim of this work consisted of the in vitro antifungal evaluation of different extracts obtained from the aerial parts of *P. acuminata* and the isolation and identification of its active compounds. In addition, an ex-vivo test was carried out using a phytochemically characterized extract on post-harvested oranges and peaches, and finally the extract cytotoxicity was compared with that of the commercial fungicides that are being used. From the results, we could conclude that the fungicidal effect of the characterized extract obtained from *P. acuminata* did not show statistically significant differences with the commercial fungicides but it displayed lower phyto-toxicity than imazalil (Imz) and carbendazim (Cbz).

## 2. Results

### 2.1. In Vitro Antifungal Activities of P. acuminata Extracts and Compounds Isolated from Them against the Selected Phytopathogenic Fungi

The three extracts obtained from dried-leaves of *P. acuminata* were evaluated for their antifungal activities against *P. digitatum*, *P. italicum,* and *M. fructicola* using the micro broth dilution assay (Table 1). Figure 1 shows a picture of the plant species in its natural environment (A) and light microscopic photographs of the three phytopathogens under study (B–D).

In addition, the three pure compounds isolated from the most active extract were assayed against the same fungal panel and the Minimum Inhibitory Concentrations (MIC) and the Minimum Fungicidal Concentrations (MFC) of all the extracts and compounds are shown in Table 1. Chemical structures of pure compounds are depicted in Figure 2.

Among the three phytopathogenic fungi selected for this work, the most sensitive to the fungicides evaluated here resulted to be *M. fructicola*. In contrast, both strains of *Penicillium* displayed higher MICs and MFCs values for all the samples evaluated, indicating that these microorganisms are very difficult to inhibit or kill (Table 1).

The hexanic extract showed moderate antifungal activity against the fungal panel, being remarkable the MIC and MFC displayed for *M. fructicola* (MIC 125 and MFC 250 µg/mL, respectively). Ethyl acetate extract was the most active with MICs and MFCs between 62.5 and 500 µg/mL, *P. italicum* being the most resistant microorganism. In contrast, methanolic extract was completely inactive for all the fungi tested here (Table 1).

Regarding the pure compounds, polygodial (**1**) showed the highest activity against the three phytopathogenic fungi, displaying MICs and MFCs from 31.25 µg/mL for *M. fructicola* to 250 µg/mL for *P. italicum*. Moreover, *P. digitatum* resulted moderately sensitive to this compound with MIC and MFC = 125 µg/mL. The sesquiterpene alcohol drimenol (**2**) and the sesquiterpene lactone confertifolin (**3**) were completely inactive against this fungi panel.

### 2.2. Detection of the Bioactive Compound in the Extracts by Bioautography

The active extracts (hexanic and ethyl acetate), in addition with the pure compound **1** as a bioactive marker, were developed on a thin layer chromatography (TLC) plate and bioautographed using *M. fructicola* as test microorganism. The same TLC plate was performed without adding the fungal inoculum but exposed to UV lights of 254 and 365 nm and sprayed with *p*-anisaldehide sulphuric to chemically detect the presence of compound **1** in both extracts. Results showed that polygodial was present in the hexanic, as well as in the ethyl acetate extracts but in the latter, its concentration was higher (Figure 3). Therefore, it was corroborated that the bioactivity observed in these extracts is mainly attributed to the content of this active compound.

### 2.3. Quantification of the Bioactive Compound in the EtOAc Extract of P. acuminata

Taking into account the variability of bioactive compounds concentrations in the complex mixture constituted by the extract and with the aim of characterizing the concentration of polygodial in the extract that was used to perform these assays; GC-EM techniques were applied. The calibration function was obtained by two-fold dilution solutions (from 500 µg/mL to 125 µg/mL) of pure compound correlated with the parameter Total Ion Chromatogram (TIC) area. Table 2 shows the values obtained for the three points of dilutions and the correspondent TIC areas as well as for the extract solution of 1000 µg/mL.

Validation parameters for the quantitative assessment of polygodial are depicted in Table 3 and Table 4, indicating linear regression data, Limit of Detection (LOD) and Limit of Quantification (LOQ), as well as intra- and inter-day precision of polygodial detection.

Results showed that the concentration of the bioactive compound polygodial in the extract used to carry out the assays for the control of the corresponding disease on fruits was 130 µg/mg of dry extract. Moreover, the same extract was used for the cytotoxicity evaluation and therefore, the same concentration of the bioactive compound determined the results of cellular viability.

### 2.4. Effect of Etoac Extract of P. acuminata for the Control of Citrus Green Mold and Peach Brown Rot on Fresh Fruits

The effect of the most active extract of *P. acuminata* was examined ex-vivo on fresh harvested oranges cv. ‘Salustiana’ infected with *P. digitatum*, where the wound-inoculated fruits were treated with the bioactive extract by dipping in a 250 μg/mL solution and using Imz (7.8 μg/mL) as commercial fungicide for comparison. After 14 days of the treatment applications, no significant differences on the number of fruits and disease severity were observed for both treatments (*P. acuminata* EtOAc extracts (Figure 4A) and Imz (Figure 4B)). The control fruits (those which had been only inoculated with the pathogen, but not treated with any fungicide (Figure 4C)) showed noticeable symptoms of the disease caused by the pathogen (see the number of sporulation index in each fruit which is described in Section 4.4.3). Both treatments (*P. acuminata* EtOAc extract and Imz) reduced significantly (*p* = 0.03) green mold sporulation index with respect to the control oranges, showing no significant differences between them.

A similar trial was also carried out on fresh peaches cv. ‘Red Globe’ but infected with the phytopathogen *M. fructicola*. The commercial product Cbz was applied as negative control for the determination of the sporulation index of infected fruits (see Section 4.4.3). As an alternative treatment, *P. acuminata* EtOAc extract was applied to wound-inoculated peaches by immersion in a 62.5 μg/mL solution of the extract. A control group of fruits without antifungal agents was also used. After 10 days of applying the treatments, it was observed that the untreated group exhibited severe symptoms of the fungal disease, which led to discard the whole lot of fruits. Additionally, the set of fruits treated with Cbz or *P. acuminata* EtOAc extract indicated clearly that both of them had protected the fruits by significantly reducing the brown rot sporulation index (Figure 5A–C). In addition, examination of the set of peaches treated with *P. acuminata* EtOAc extract revealed that less fruits showed symptoms of fungal infection while those exposed to commercial Cbz appeared to be infected in a greater degree. Furthermore, despite no significant difference was observed between the fruits treated with the commercial fungicide and those treated with the alternative one proposed in this work, the results suggested that the latter could be even more powerful than the commercial agent. This requires further investigations.

### 2.5. Cytotoxicity Evaluation of P. acuminata EtOAc Extract Compared with Commercial Fungicides

As shown in Figure 6, the viability of Huh7 cells was 76.27 ± 2.86%, 45.74 ± 3.08% and 41.74 ± 2.07% in the presence of *P. acuminata* EtOAc extract, Cbz and Imz, respectively, at their MICs values (2-fold dilutions). This means that the extract seems to be less cytotoxic than commercial antifungals at the concentrations evaluated by this method, since the higher cellular viability values means lower cytotoxicity. Notice that the abscissa axis shows the serial dilutions from the most concentrated sample solutions (dilutions 1), which are twice as concentrated as their MIC values (2-fold dilutions), and so on. Therefore, 4-fold dilutions are represented as the most dilute solutions of each sample, and they present the highest cellular viability.

## 3. Discussion

In this work, we explored the in vitro antifungal activity of different extracts obtained from the leaves of *P. acuminata*. The ethyl acetate extract showed the highest bioactivity against a panel of three fruit phytopathogenic fungi including: *P. digitatum*, *P. italicum,* and *M. fructicola*. The sesquiterpenes isolated from the extract were also evaluated against these fungal strains, demonstrating that the dialdehyde polygodial was the responsible for this activity. In order to encourage the use of the extract rather than the pure compound, we displayed ex vivo assays using fresh oranges and peaches inoculated with *P. digitatum* and *M. fructicola*, respectively, and subsequently treated by immersion with an extract solution of 250 and 62.5 µg/mL. There were no statistically significant differences between the treatments with the commercial fungicides and the extract over the control of both fruit rots. The concentration of the active compound present in the extract used on fruit experiments was determined by GC-MS, resulting in 130 µg/mg of dry extract. Finally, cytotoxicity evaluation against Huh7 cells showed that *P. acuminata* extract was less cytotoxic than the commercial fungicides at the assayed concentrations.

The genus *Polygonum* (Polygonaceae) has been widely studied from a phytochemical point of view and by its interesting bioactivities. The genus includes approximately 300 species that are widely distributed in temperate regions of the world, but since most of them grow in China, many reviews describing the 113 species used in the traditional Chinese medicine appear in the literature [22]. Phenolic compounds from *P. cuspidatum* and *P. bistorta* demonstrated cytotoxicity against different type of cancer cells while their stilbenes and anthraquinones are potent anti-oxidants. These last type of compounds, but isolated from *P. aviculare* and *P. glabrum*, showed a broad antibacterial range (Gram-positive and Gram-negative), as well as antifungal properties [22]. A methanolic extract of *P. maritum* collected from the Argelian coast proved to be active against *Bacillus cereus*, *B. subtilis,* and *Staphylococcus aureus*, probably due to its high content of phenolic compounds. Moreover, the crude methanolic extract obtained from *P. viscosum* leaves was as anthelminthic as standard drugs [22]. Seeds of *P. equisetiforme* collected from different regions of Tunisia were characterized for the presence of unsaturated fatty acids, flavonoids and tannins, which were strong 2,2-diphenyl-1-picrylhydracil (DPPH) scavengers, highlighting the nutritive value of these seeds as antioxidants and the source of bioactive compounds [23].

The native Argentinian herb *Persicaria acuminata syn*. *Polygonum acuminatum* had been reported previously for its antifungal properties against the human fungal pathogens *Candida albicans*, *C. tropicalis*, *Cryptococcus neoformans,* and the dermatophytes *Trichophyton mentagrophytes*, *T. rubrum,* and *Microsporum canis* with MICs and MFCs between 15.6 and 62.5 µg/mL according the Clinical & Laboratory Standards Institute techniques [24]. In the same article, a seasonal variation study demonstrated that dichloromethane extracts of *P. acuminata* leaves collected in autumn had the highest content of polygodial and the lowest MICs against the fungi tested, being the most suitable for treating conditions related to fungal infections in humans [19]. Based on this knowledge, we directed our new lines of investigations towards the control of fruit phytopathogenic fungi with the aim of applying active extracts or compounds in fresh fruit experiments. In this work, we demonstrated that EtOAc extract of *P. acuminata* was the most active against the phytopathogens *P. digitatum*, *P. italicum*, and *M. fructicola* (MICs and MFCs between 62.5 and 1000 µg/mL), being much less active than against human pathogens, as reported by Derita et al. (2009) [19]. Moreover, methanolic extract was inactive in accordance with the previous results [19]. The three sesquiterpenes (polygodial, drimenol, and confertifolin) were isolated from the most active extract, but in contrast with the previous work [19,21] only polygodial showed activity against the phytopathogens evaluated here, demonstrating by bioautography that this compound was the bioactive marker for the EtOAc extract. Additionally, its anti-phytopathogenic activities (MICs and MFCs between 31.25 and 250 µg/mL) were lower than those displayed against human pathogens (MICs and MFCs between 3.9 and 62.5 µg/mL), indicating the difficulty of the eradication of this type of filamentous fungi.

Many sesquiterpenes obtained from another plant source (*Drimys winteri* Forst) have been reported against phytopathogens in vitro [6,25]. Particularly, Montenegro et al. (2018) reported high MICs values (from 16 to 32 µg/mL) for the sesquiterpenes polygodial, drimenol, isonordrimenone, and nordrimenone against the causative agents of bacterial canker (*Clavibacter michiganensis*) and bacterial speck (*Pseudomonas syringae pv*. *tomato*) on tomato crops [25]. Moreover, the same compounds showed important antimycotic activity against *Fusarium oxysporum* and *Phytophthora* spp. (tomato pathogens) with MIC = 64 µg/mL [25].

The high antimicrobial activity of the sesquiterpenes reported by several authors led to a Structure-Activity Relationships analysis of a series of 17 drimanes, supported by conformational and electronic studies, which allowed to show that the Δ^7,8^-double bond would be the main structural feature related to the antifungal activity [26]. These results could support our findings related to the activity of polygodial and its presence in the EtOAc extract of *P. acuminata* as the bioactive marker.

Inspired by previous studies over fresh fruits performed by other authors [27] and adapting the methodologies to our experiments, we successfully evaluated the action of *P. acuminata* extract containing 0.13 mg of polygodial/mg of dry extract on oranges and peaches inoculated with *P. digitatum* and *M. fructicola*, respectively. Oranges exposed to commercial Imz were visibly infected at the same level as those treated with the extract, showing no statistically significant differences between both; furthermore, the infected area seemed to be smaller in the oranges treated with the extract than in those exposed to the commercial product. Moreover, the incidence of brown rot disease on fresh ‘Red Globe‘ peaches were compared with the application of the commercial product Cbz, through the determination of the sporulation index of the infected fruits. To that end, the extract was applied to wound-inoculated peaches by immersion in a 62.5 μg/mL solution and no statistically significant differences were appreciated between both treatments.

Further experiments that use more fruits and formulation adding adjuvants to improve the efficacy of the extract are proposed to be carried out. Moreover, the application of *P. acuminata* EtOAc extract on other types of fruit would be interesting to assay.

The experimental results suggest that the potencies of *P. acuminata* EtOAc extract, chemically characterized by its bioactive marker, and the commercial antifungal agents against the tested phytopathogens are not too different. Remarkably, primary cytotoxic experiments demonstrated the lower toxicity of the extract compared with the commercial fungicides at the tested concentrations. After all these findings, we suggest that this natural product should be considered as an excellent candidate for further development to treat fungal fruit diseases.

## 4. Materials and Methods 

### 4.1. Plant Material and Preparation of the Extracts

Leaves from the adult stage of *Persicaria acuminata* were collected in March 2016 around the city of Puerto Gaboto (Santa Fe province, Argentina) in one of the arms of the Coronda River. The wild plants were identified by Professor Susana Gattuso from the National University of Rosario (UNR) and a *voucher specimen* was deposited at the Herbarium of the Vegetal Biology Area of UNR (uipacha 531-(2000)-Rosario, Argentina, code: UNR 1672).

Air-dried leaves (100 g) were mechanically powdered. Stems and roots were not used because it has been previously demonstrated that the antifungal compounds are only present in leaves [19]. The material was successively macerated (3 × 24 h each) with hexane (Hex), ethyl acetate (EtOAc) and methanol (MeOH), with mechanical stirring. After filtration and evaporation, Hex (1.5 g), EtOAc (2.1 g), and MeOH (2.7 g) crude extracts were obtained and stored in a freezer at −4 °C.

### 4.2. Isolation and Chemical Characterization of Natural Compounds **1**–**3**


Compounds **1**–**3** were isolated from EtOAc extracts of leaves of *P. acuminata.* The isolation of the pure compounds was performed according to previously reported procedures [19,20,21]. Compounds **1**–**3** were identified by micromelting point, optical rotation, and spectroscopic data, including ^1^H and ^13^C Nuclear Magnetic Resonance and were compared with authentic samples obtained previously in our laboratory [19,28] for polygodial (**1**) or with the literature data, for drimenol (**2**) and confertifolin (**3**) [29,30]. Chemical compound descriptions and spectroscopic data are presented in the Appendix A).

### 4.3. Quantification of Polygodial in the Most Active Extract

The EtOAc extract was submitted to GC-MS using a Turbo Mass Perkin Elmer chromatograph, equipped with a fused silica gel column (SE-30 25 m × 0.22 mm ID) with He as carrier gas, coupled to a mass selective detector, film 0.25 μm, ionization energy 70 eV with a temperature programmer of 70–200 °C at 10 °C/min; total time 30 min. Polygodial (**1**) was identified by comparison of its retention time (17.81–17.82 min) and the Mass Spectroscopy spectrum with an authentic sample obtained from our previous work [19]. Chromatograms and MS spectra are presented in the Appendix A).

Quantification of polygodial content was carried out using Gas Chromatography–Mass Spectrometry that was first validated following the International Council for Harmonisation (ICH) guidelines (1996) [31]. Stock solutions were diluted to appropriate concentrations in order to construct a calibration curve of polygodial and calculate relative response factors.

*Linearity and calibration curve*: Linearity of polygodial calibration curve was established by calculating the slope, intercepts and *R*^2^ coefficient. The regression equation (y = 91400x − 113.2) and *R*^2^ (0.996) showed good linearity response in the range 1000 to 31.2 (µg/mL). LOD and LOQ were found to be 0.008 and 0.023 µg/mL.

*Precision*: Intra- and inter-day variability test was determined for three times within 1 day and 3 consecutive days at three different concentrations (low, medium, and high), respectively. Variations were expressed by the percentage of relative standard deviation (% RSD), confirming the precision of the proposed method.

*Accuracy*: In total, three concentrations (low, medium and high) of pre-analyzed sample solutions were spiked with known quantities of the standard and injected in triplicate to perform recovery studies. The percentage recovery for polygodial was between 97.5 and 99.6% (RSD < 4%, *n* = 3), confirming the accuracy of the proposed method.

### 4.4. Antifungal Studies

#### 4.4.1. Microorganisms and Media

Monosporic strains of each fungus were obtained from fruits that exhibited the corresponding disease symptoms, and were morphologically characterized by the Mycology Reference Center (CEREMIC-CCC, Rosario, Argentina) and the National Institute of Agricultural Technology (INTA, EE San Pedro, Argentina). The strains of *P. digitatum* CCC-102, *P. italicum* CCC-101 and *M. fructicola* INTA-SP345, were grown on Petri dishes filled with the culture medium Potato Dextrose Agar (PDA) and incubated during 7 to 10 days at 20–25 °C (as needed for the suitable growth of each one). The inoculum of spore suspensions were obtained according to the CLSI reported procedures and adjusted to 1 × 10^4^ Colony Forming Units (CFU)/mL [24].

#### 4.4.2. In Vitro Susceptibility Tests

The MIC values were determined by using broth microdilution techniques according to the CLSI guidelines [24] for filamentous fungi (document M38). Microtiter trays were incubated at 20–25 °C in a moist dark chamber, and MIC values were visually recorded at a time according to the growth control of each fungus evaluated. For the assay, solutions of each extract/compound (50 mg/mL or 12.5 mg/mL, respectively) were prepared in Dimethylsulfoxide. Aliquots (40 μL) of these were diluted in the liquid media Sabouraud 2× (960 μL) to obtain stock solutions that were serially diluted from 500 to 7.9 μg/mL (final volume = 100 μL) in the corresponding wells of a microtiter plate. A volume of fungal suspension (100 μL) was added to each well, except for the sterility control, where sterile water was used instead. The commercial antifungal agents Imz and Cbz were used as positive controls. The MIC endpoints were defined as the lowest concentration of extract or compound, visually resulting in the total inhibition of the fungal growth compared to the growth in the control wells containing no antifungal agent.

The Minimum Fungicidal Concentration (MFC) was defined as the lowest concentration of extract or compound that fully killed the fungi. The MFC values were determined after assessing the corresponding MIC data, by transferring sample aliquots (5 μL) from each clear well of the microtiter tray onto a 150-mm PDA plate. The inoculated plates were incubated at 20–25 °C and the MFC values were recorded after 7 to 10 days, according to the corresponding growth control. The evaluated samples were considered inactive when MIC or MFC resulted higher than 1000 µg/mL for extracts or higher than 250 µg/mL for pure compounds.

For bioautography, chromatograms performed in TLC plates using the mixture Hexane:EtOAc (7:3) as mobile phase were placed in sterile Petri dishes with cover. Potato Dextrose growth media with 0.6% agar and 0.02% phenol red containing an inoculum of *M. fructicola* of 1 × 10^4^ CFU/mL, quantified according to reported procedures [32], was distributed over the TLC plate (1 mL/cm^2^) containing the samples (15 µg of the hexanic and EtOAc extracts and 5 µg of the pure compound). After solidification of the media, the TLC plates were incubated for 6 days at 20 °C. Subsequently, bioautograms were sprayed with an aqueous solution (1 mg/mL) of methylthiazolyltetrazolium chloride (MTT) and incubated for another 3 h at 20 °C. Dark yellow inhibition zones appeared against a dark brown background [33].

#### 4.4.3. Ex Vivo Antifungal Assays Using *P. acuminata* Chemically Characterized Extract on Wounded Fruits

Oranges (cv. ‘Salustiana’) and peaches (cv. ‘Red Globe’) were harvested from the Experimental Field of Intensive and Forestry Crops (*Facultad de Ciencias Agrarias, Universidad Nacional de Litoral*) at the mature stage and sorted based on size and absence of physical injuries or disease infection. No post-harvest commercial fungicide treatments had been applied before the fruits were collected. The oranges and peaches were cleaned and surface-disinfected with 2% (*w/v*) sodium hypochlorite for 3 min, rinsed with tap water, and then air-dried. Once treated with the fungicides proposed in this work, the fruits were put in 3 plastic boxes of 300 mm × 500 mm × 100 mm covered in the bottom with filter papers embedded with 25 mL of sterile water in order to maintain a high relative humidity (90–95%). Each plastic box contained the fruit units that had been submitted to the same treatment: negative control using commercial fungicide, positive control using sterile water, or *P. acuminata* extract solution as the alternative fungicide.

The fruits were stored at 20 °C for 14 days (oranges) or 10 days (peaches) in accordance with the disease evolution of the control sets. After storage, the degree of *P. digitatum* (for oranges) or *M. fructicola* (for peaches) sporulation on the surface of decayed fruits was evaluated on a 0 to 4 scale (sporulation index). In this scale, the incidence and severity of the disease were visually quantified assuming the following values: 0 (negligible sporulation); 1 (fruits with lesions up to 10% of its surface); 2 (fruits with injuries between 10 and 30% of its surface); 3 (fruits with lesions between 30 and 50% of its surface) and 4 (refers to a dense fungal sporulation over the entire fruit that infected in more than 50% of its surface). The index value for each fruit was treated as a replicate, and each treatment mean was subjected to statistical analysis [27,34,35].

For the trial with oranges, each fruit was wounded (3 mm deep and 3 mm wide) with a sterile nail in the top of the fruit and 10 µL of a conidial suspension (10^5^ CFU/mL) of *P. digitatum* was placed into each wound. Once inoculated, the fruits were randomly distributed into 3 groups with 10 units each: one group was submitted to a treatment with EtOAc extract of *P. acuminata* (250 μg/mL) according to the results obtained by the in vitro antifungal assay; while the other groups were used as negative control (sterile water treatment) and positive control employing commercial Imz at 7.8 μg/mL, according also to the in vitro assay result. The treatments were carried out after 2 h of inoculation, by immersion during 3 s of each fruit into a beaker with the corresponding solutions mentioned above.

For the trial with peaches, each fruit was wounded with a sterile tip in the upper zone and 10 µL of a conidial suspension (10^5^ CFU/mL) of *M. fructicola* was placed into each wound. Once inoculated, the fruits were randomly distributed into 3 groups with 10 units each: one group was submitted to a treatment with EtOAc extract of *P. acuminata* (62.5 μg/mL) according to the results obtained by the in vitro antifungal assay; while the other groups were used as negative control (sterile water treatment) and positive control employing commercial Cbz at 1.9 μg/mL, according also to the in vitro assay result. The treatments were carried out after 4 h of inoculation, by immersion during 3 s of each fruit into a beaker with the corresponding solutions mentioned above.

Statistical Analysis: Experimental data were analyzed statistically by one-way ANOVA followed by Tukey’s multiple comparison test (α = 0.05) using the GraphPad Prism 7.0 software.

### 4.5. Cell Viability Assay

Human hepatoma (Huh7) cells were treated for 24 h with different concentrations of EtOAc extract of *P. acuminata*, Imz and Cbz (2×, 1×, ½× and ¼× MIC) and the cell viability was estimated by the MTT assay [36]. Experiments were done by triplicate and mean followed by standard deviation were calculated. As a positive control (100% death), 10% DMSO was used as the starting dilution, and the reading for all the dilutions was considered 0% viability. As a negative control (0% death), 0.1% DMSO was used, and the reading for all the dilutions was considered 100% viability.

## Figures and Tables

**Figure 1 plants-10-00425-f001:**
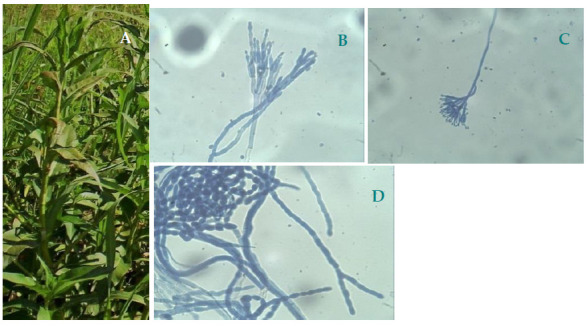
(**A**) Picture of the species *Persicaria acuminata* in its natural environment. Light microscopic photos (40×) of the phytopathogens *Penicillium digitatum* (**B**), *Penicillium italicum* (**C**), and *Monilinia fructicola* (**D**). Both species of *Penicillium* show their characteristic conidiophores, finger-shaped phialides and apical spores. Characteristic chains of lemon-shaped spores are observed for *M. fructicola*.

**Figure 2 plants-10-00425-f002:**
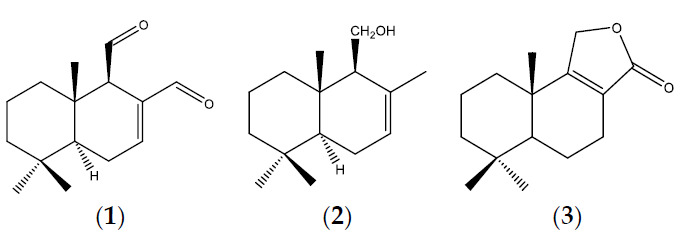
Chemical structures of the compounds isolated from Ethyl Acetate extract of leaves of *Persicaria acuminata*. (**1**) Polygodial; (**2**) Drimenol; (**3**) Confertifolin.

**Figure 3 plants-10-00425-f003:**
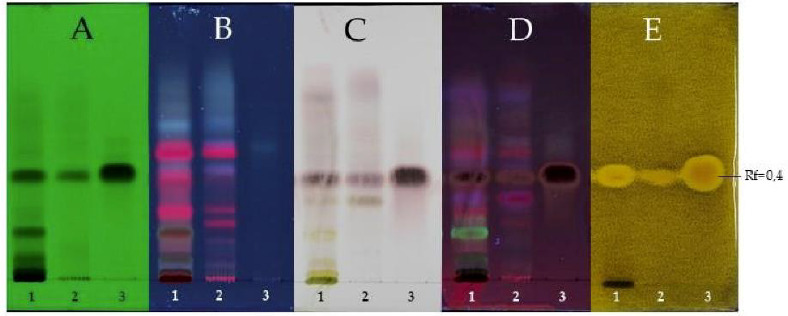
Thin Layer Chromatography developed for EtOAc extract of *P. acuminata* (**1**), hexanic extract of *P. acuminata* (**2**) and the pure compound polygodial (**3**: Rf = 0.4). (**A**) exposed to UV light 254 nm; (**B**) exposed to UV light 365 nm; (**C**) sprayed with *p*-anisaldehide sulphuric; (**D**) sprayed wih *p*-anisaldehide sulphuric and exposed to UV light 365 nm; (**E**) bioautography using *M. fructicola* as tested microorganism. Mobile phase: Hexane:EtOAc (7:3).

**Figure 4 plants-10-00425-f004:**
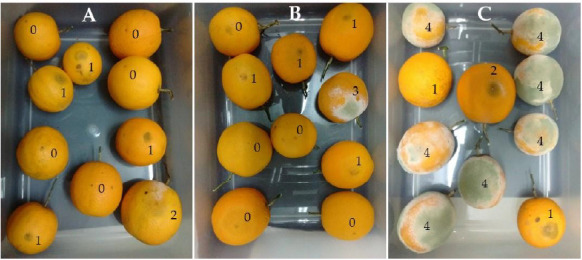
Pictures showing P. digitatum sporulation index (0 to 4 scale, in which the value 0 was assigned to negligible sporulation and 4 referred to a dense fungal sporulation over the entire fruit) on wound-inoculated oranges treated with *Persicaria acuminata* EtOAc extract (**A**), commercial imazalil fungicide (Imz), (**B**) and the control set, (**C**) without any treatment.

**Figure 5 plants-10-00425-f005:**
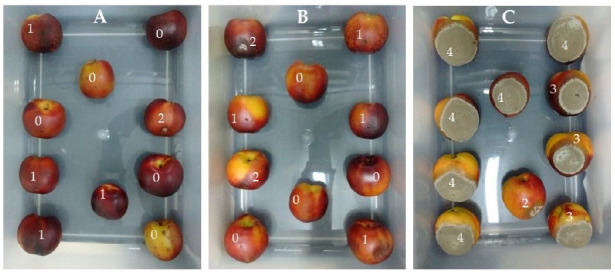
Pictures showing M. fructicola sporulation index (0 to 4 scale, in which the value 0 was assigned to negligible sporulation and 4 referred to a dense fungal sporulation over the entire fruit) on wound-inoculated peaches treated with *Persicaria acuminata* EtOAc extract (**A**), commercial Carbendazim (**B**) and the control set, (**C**) without any treatment.

**Figure 6 plants-10-00425-f006:**
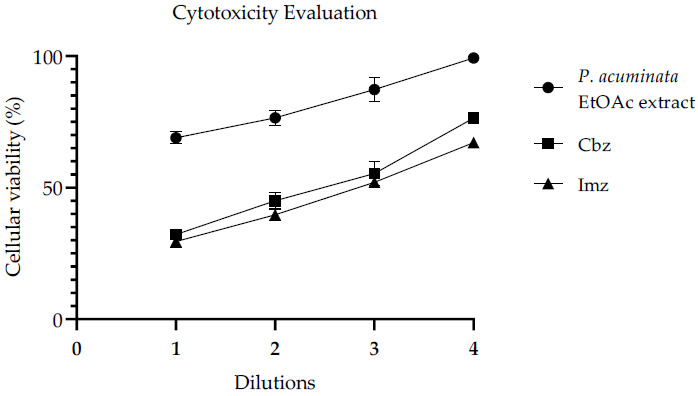
Cellular viability of Huh7 cells in the presence of *P. acuminata* EtOAc extract and commercial fungicides Cbz and Imz at different concentrations: Dilutions 1, 2, 3, and 4 correspond to 2×, 1×, ½×, and ¼× MIC, respectively. Values are expressed as mean ± SD, determined by triplicate.

**Table 1 plants-10-00425-t001:** Minimum Inhibitory Concentrations (MIC) and Minimum Fungicidal Concentrations (MFC) (µg/mL) of extracts and pure compounds isolated from Persicaria acuminata against the selected post-harvest fruits pathogens.

Sample	MICs/MFCs (µg/mL) ^1^		
	*P. digitatum*	*P. italicum*	*M. fructicola*
Hexanic extract	500/1000	500/1000	125/250
EtOAc extract	250/250	500/500	62.5/62.5
MeOH extract	I/I	I/I	I/I
Polygodial (**1**)	125/125	250/250	31.25/31.25
Drimenol (**2**)	I/I	I/I	I/I
Confertifolin (**3**)	I/I	I/I	I/I
Imazalil (Imz)	7.8/7.8	15.6/15.6	0.97/0.97
Carbendazim (Cbz)	15.6/15.6	15.6/15.6	1.9/1.9

^1^*Penicillium digitatum* CCC-102; *Penicillium italicum* CCC-101; *Monilinia fructicola* INTA-SP345. The commercial fungicides Imazalil (Imz) and Carbendazim (Cbz) were used as positive controls. I: inactive (MIC or MFC > 1000 µg/mL for extracts or > 250 µg/mL for pure compounds).

**Table 2 plants-10-00425-t002:** Total Ion Chromatogram areas obtained for the two-fold diluted solutions of pure polygodial (from 500 to 125 µg/mL) and the extract solution (1000 µg/mL) using Gas Chromatography-Mass Spectroscopy. Experiments were done by triplicate.

Pure Polygodial Concentration (µg/mL)	TIC Areas
500	1.61 × 105
250	7.97 × 104
125	1.34 × 104
**EtOAc extract of *P. acuminata* (1000 µg/mL)**	**TIC area**
130	5.22 × 104

**Table 3 plants-10-00425-t003:** Linear regression data, limit of detection (LOD) and limit of quantification (LOQ) of polygodial detection.

	Linear Regression Data			Limits	
	Regression Equation	*R* ^2^	Linear Range (µg/mL)	LOD (µg/mL)	LOQ (µg/mL)
Polygodial	y= 91400x − 113.2	0.996	1000–31.2	0.008	0.023

**Table 4 plants-10-00425-t004:** Intra- and inter-day precision of polygodial.

	Precision		
	Concentration (µg/mL)	Intra-Day (*n* = 3), RDS %	Inter-Day (*n* = 3) RDS %
Polygodial	28.00	2.1	2.4
	14.00	2.1	2.0
	1.00	2	2.3

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
