# Peer review of "Botanical Control of Citrus Green Mold and Peach Brown Rot on Fruits Assays Using a Persicaria acuminata Phytochemically Characterized Extract"

_plants, 2021, doi:10.3390/plants10030425_

Round 1

Reviewer 1 Report

Dear colleague (s),

This review concerns “Botanical control of citrus green mold and peach brown rot on fruits assays using a Persicaria acuminata phytochemically characterized extract”, by Melina G. Di Liberto, Gisela M. Seimandi, Laura N. Fernández, Verónica E. Ruiz, Laura A. Svetaz and Marcos G. Derita.  As detailed experiments I recommend it for an international audience in this journal.

However, several points have to be precised and a minor revision is requested. Please notice that the two major points of my comments (at the beginning) are very important (mandatory…) for a suitable value and understanding of the article, especially for non-specialists in order to bring a broader audience in this journal.

I deeply hope to see this article published,

The two major point are:

  • As this paper concerns quite a rare and very interesting taxon of Argentina, a plate with the photo of this plant is highly required; moreover this plate should also contain photos (with light microscopy or others) of the fungi studied (detailing in the captions in a few words their (morphological) specificities), largely studied but unfortunately not visible in the article; in this respect “cellular viability” is used in the text, but in the present manuscript no cells are visible in the illustrations, which is quite frustrating for all these very interesting set of observations and measurements.

  • Checking briefly in the word of science WOS with key-words and topics like “Persicaria, Polygonum, sesquitepernes, Polygonaceae, polygodial,  antifung* sensu lato…, much more articles (very recent from 2020 and even 2021, or “older”) appear and they should be once more selected and used in their details and values provided  (not just citing them)  to increase (much more, although it is already quite well-discussed…) the quality of the discussion for an international audience in this journal.

The minor points are:

1 in the introduction, p. 2 precise or reword “conditions conducive”;

2 in the introduction, in the paragraph « Persicaria acuminata (Kunth) M.Gómez syn. Polygonum…”, put also the argentinian word for “big catay”; if this Argentinian word relates to antifungi property of this taxon, please give some details which are always welcome for botanists (I am a botanist myself…);

3 in the results, precise the reason why you use “dried leaves” (and not stems, roots…), or precise this in “material and methods” part;

4 in point 2.1, just below Figure 1, reword or precise “indicating that they are types of microorganisms that are more difficult to inhibit or kill”;

5 in the title of 2.2, is “bioatography” the proper word ?; this word also appears in the first paragraph of the discussion;

6 for figures 3 and 4: indicate more in details in the captions (as you did in the text but more shortly)  the meaning of "o, 1 2 3 4" in A B C boxes;  it is also not clear if in the diagrams "a" and "b" are necessary for the understanding as the differences between the values are quite clear in the boxplots without these letters; moreover put the latin name of the taxa in full letters, it is more rapidly understandable in captions;

7 in point 2.4, in the paragraph “An analogous test was also carried…”,  precise or reword “After 10 days, it was observed that the control…”;

8 in point 2.5, precise or reword “since the higher cellular viability the lower cytotoxicity.”;

9 in the pdf version I am reviewing, figure 5 contains some overlappings of words or symbols, please check this with the editor;

10 in the discussion, at the beginning of this part use full latin letters for “P. acuminata”, it is much easier to read; also do this in 4.1;

11 in this part or in the methods, precise the meaning of “active and unactive” extract, it will be more readable for a non-specialist;

12 in the material and methods section, the different parts should be renumbered as in the present state they are all numbered “4.1”;

13 in the material and methods part, precise and argument “how many leaves were used and at which stage (adult, young... why ? I suppose that there is a chemical-physiological reason but it has to be enhanced);

14 at the end of the “first” 4.1 (Plant material and preparation of the extracts), after “crude extracts were obtained”, precise how the material was stored;

15 in the “Ex vivo antifungal assay in wounded fruits” section, precise somewhere the number of all fruits and boxes used; use eventually these values to discuss the accuracy of the results;

16 in the “Test development and evaluation” part, a percentage (even approximate as I understand it is difficult to evaluate very precisely) for each of the 0-4 degrees of sporulation would be welcome;

17 in the supplemental material pdf file (named “plants-1074176-non-published”), precise if all diagrams shown are examples of results, moreover in order to save space in the journal the size of all these diagrams should be reduced;

18 as the last sentence of the conclusion (“After all these findings,…) seems to be very important for the future of this research, it should be much more enhanced in the abstract.

Author Response

Response to Reviewer 1 Comments

Point 1 (mandatory): As this paper concerns quite a rare and very interesting taxon of Argentina, a plate with the photo of this plant is highly required; moreover this plate should also contain photos (with light microscopy or others) of the fungi studied (detailing in the captions in a few words their (morphological) specificities, largely studied but unfortunately not visible in the article; in this respect “cellular viability” is used in the text, but in the present manuscript no cells are visible in the illustrations, which is quite frustrating for all these very interesting set of observations and measurements.

Response 1: Figure 1 was added. This plate contains a picture of the plant species under study and a photo taken by light microscopy of each phytopathogen evaluated. Moreover, fungal morphological specificities are detailed in the caption of Figure 1.

Point 2 (mandatory): Checking briefly in the word of science WOS with key-words and topics like “Persicaria, Polygonum, sesquitepernes, Polygonaceae, polygodial,  antifung* sensu lato…, much more articles (very recent from 2020 and even 2021, or “older”) appear and they should be once more selected and used in their details and values provided  (not just citing them)  to increase (much more, although it is already quite well-discussed…) the quality of the discussion for an international audience in this journal.

Response 2:

The quality of the discussion for an international audience was improved by commenting recent reviews regarding the phytochemistry and bioactivities of the genus at the beginning of discussion section. 

The minor points are:

  • In the introduction, p. 2 precise or reword “conditions conducive”

Response: It was changed to “predisposing conditions”

  • In the introduction, in the paragraph « Persicaria acuminata (Kunth) M.Gómez syn. Polygonum…”, put also the Argentinian word for “big catay”; if this Argentinian word relates to antifungal property of this taxon, please give some details which are always welcome for botanists (I am a botanist myself…)

Response: “big catay” was changed to the Argentinian word “catay grande”. Unfortunately, this word do not relate to antifungal properties.

  • In the results, precise the reason why you use “dried leaves” (and not stems, roots…), or precise this in “material and methods” part.

Response: It was specified in material and methods section: “Stems and roots were not used because it has been previously demonstrated that the antifungal compounds are only present in leaves [19]”.

  • In point 2.1, just below Figure 1, reword or precise “indicating that they are types of microorganisms that are more difficult to inhibit or kill”

Response: it was changed by “indicating that these microorganisms are very difficult to inhibit or kill”

  • In the title of 2.2, is “bioatography” the proper word ?; this word also appears in the first paragraph of the discussion

Response: bioautography is the proper word for this bioassay. It was corrected.

  • For figures 3 and 4: indicate more in details in the captions (as you did in the text but more shortly) the meaning of "0, 1 2 3 4" in A B C boxes; it is also not clear if in the diagrams "a" and "b" are necessary for the understanding as the differences between the values are quite clear in the boxplots without these letters; moreover put the Latin name of the taxa in full letters, it is more rapidly understandable in captions.

Response: In the captions of figures 3 and 4 (now 4 and 5) it was indicated the meaning of 0 to 4 and the Latin name of the species was put in full letter. The diagrams were deleted.

  • In point 2.4, in the paragraph “An analogous test was also carried…”, precise or reword “After 10 days, it was observed that the control…”

Response: It was reworded to “after 10 days of applying the treatments”.

  • In point 2.5, precise or reword “since the higher cellular viability the lower cytotoxicity.”

Response: It was reworded to “since the higher cellular viability values means lower cytotoxicity”

  • In the pdf version I am reviewing, figure 5 contains some overlapping of words or symbols, please check this with the editor.

Response: it was solved.

  • In the discussion, at the beginning of this part use full Latin letters for “ acuminata”, it is much easier to read; also do this in 4.1.

Response: It was used full Latin letters.

  • In this part or in the methods, precise the meaning of “active and inactive” extract, it will be more readable for a non-specialist.

Response: It was clarified in methods, section 4.4.2.

  • In the material and methods section, the different parts should be renumbered as in the present state they are all numbered “4.1”

Response: They were correctly renumbered.

  • In the material and methods part, precise and argument “how many leaves were used and at which stage (adult, young... why? I suppose that there is a chemical-physiological reason but it has to be enhanced).

Response: It was specified that leaves at adult stage were used. Unfortunately, the number of leaves used was not counted, it was determined by weight.

  • At the end of the “first” 4.1 (Plant material and preparation of the extracts), after “crude extracts were obtained”, precise how the material was stored.

Response: It was specified. They were stored in a freezer at -4ºC.

  • In the “Ex vivo antifungal assay in wounded fruits” section, precise somewhere the number of all fruits and boxes used; use eventually these values to discuss the accuracy of the results.

Response: it was specified the number of all fruits and boxes used in the corresponding section.

  • In the “Test development and evaluation” part, a percentage (even approximate as I understand it is difficult to evaluate very precisely) for each of the 0-4 degrees of sporulation would be welcome.

Response: A percentage for each of the 0-4 degrees of sporulation was specified.

  • In the supplemental material pdf file (named “plants-1074176-non-published”), precise if all diagrams shown are examples of results, moreover in order to save space in the journal the size of all these diagrams should be reduced.

Response: All the diagrams in the supplementary material correspond to the different spectra obtained for the compounds isolated in this work. It is important to place them because they ensure the identity of the compounds.

  • As the last sentence of the conclusion (“After all these findings,…) seems to be very important for the future of this research, it should be much more enhanced in the abstract.

Response: This idea was enhanced in the last sentence of the abstract.

Reviewer 2 Report

This paper entitled Botanical control of citrus green mold and peach brown rot on fruits assays using a Persicaria acuminata phytochemically characterized extract is interesting and important, however, this manuscript needs considerable revision.

Materials and Methods

Isolation and chemical characterization of natural compounds 1-3

Quantification of polygodialin the most active extract

In these sections more details are needed.

How the method was validated?

What kind of validation parameters have been checked?

How about the validation results?

What was the LOD and LOQ of the method?

Author Response

Response to Reviewer 2 Comments

Point 1:

Materials and Methods: Isolation and chemical characterization of natural compounds 1-3, Quantification of polygodial in the most active extract. In these sections more details are needed:

Response: These sections were highly improved. See the track changes in the revised version.

How the method was validated?

Response: Analytical method was first validated following the ICH guidelines (ICH 1996).

What kinds of validation parameters have been checked?

Response: Linearity, limit of detection and quantification (LOD and LOQ) and inter-day/intra-day precision were validated. Recovery was used to evaluate the accuracy of the method. Table 3 and 4 were added.

How about the validation results?

Response: The validation results were added (Table 3 and 4).

What were the LOD and LOQ of the method?

Response: They were specified in Table 3.

Round 2

Reviewer 2 Report

The revised version addresses the comments raised by the reviewers. I have no more suggestions.

Author Response

February 13th, 2021

Plants MDPI

Dear Editor,

Please find enclosed the Manuscript correction for the work entitled “Botanical control of citrus green mold and peach brown rot on fruits assays using Persicaria acuminata characterized extract, by Melina G. Di Liberto, Gisela M. Seimandi, Laura N. Fernández, Verónica E. Ruiz, Laura A. Svetaz and Marcos G. Derita, which we would like to be published in the Special Issue "Natural Products for Plant Pest and Disease Control” for publication as a Research Article.

We have carefully replied to the academic editor notes:

1) Minor modifications (mainly to the English) were done.

2) All binomial names were italicized.

3) Results section 2.4 was re-worded according to the suggestions.

3) At the start of the Discussion, a paragraph that summarized the main findings was added.

4) Further research proposal was mentioned before de final paragraph in the discussion section.

Please let me clarify some points that are also included as comments in the MS correction:

  • Regarding to fungal diseases, the negative control is the one that does not allow the disease to develop.
  • The antifungal treatments were carried out by immersion in the solutions, not by spraying.
  • Figures 4 and 5C correspond to the control fruits (those that had no received antifungal treatments); meanwhile figures 4 and 5A correspond to the fruits treated with the EtOAc extract of acuminata. Figures 4 and 5B correspond to the fruits treated with the commercial fungicide.

Thank you very much for your courtesy,

Sincerely yours,

Dr. Marcos G. Derita

Instituto de Ciencias Agropecuarias del Litoral (CONICET-UNL)

Facultad de Ciencias Bioquímicas y Farmacéuticas (UNR)

Kreder 2805, 3080 - Esperanza (Santa Fe), ARGENTINA

Tel.: 54-9341-5317769

e-mail: mderita@fbioyf.unr.edu.ar, mderita@hotmail.com
